# Structural basis for two-way communication between dynein and microtubules

Noritaka Nishida [1,3,4], Yuta Komori [2,4], Osamu Takarada[1,4], Atsushi Watanabe[1], Satoko Tamura[1], Satoshi Kubo[1], Ichio Shimada [1✉] & Masahide Kikkawa [2✉]

The movements of cytoplasmic dynein on microtubule (MT) tracks is achieved by two-way communication between the microtubule-binding domain (MTBD) and the ATPase domain via a coiled-coil stalk, but the structural basis of this communication remains elusive. Here, we regulate MTBD either in high-affinity or low-affinity states by introducing a disulfide bond to the stalk and analyze the resulting structures by NMR and cryo-EM. In the MT-unbound state, the affinity changes of MTBD are achieved by sliding of the stalk α-helix by a half-turn, which suggests that structural changes propagate from the ATPase-domain to MTBD. In addition, MT binding induces further sliding of the stalk α-helix even without the disulfide bond, suggesting how the MT-induced conformational changes propagate toward the ATPase domain. Based on differences in the MT-binding surface between the high- and low-affinity states, we propose a potential mechanism for the directional bias of dynein movement on MT tracks.

[1] Graduate School of Pharmaceutical Sciences, The University of Tokyo, 7-3-1 Hongo, Bunkyo-ku, Tokyo 113-0033, Japan. [2] Department of Cell Biology and Anatomy, Graduate School of Medicine, The University of Tokyo, 7-3-1 Hongo, Bunkyo-ku, Tokyo 113-0033, Japan. [3] Present address: Graduate School of Pharmaceutical Sciences, Chiba University, 1-8-1 Inohana, Chuo-ku, Chiba 260-8675, Japan. [4] These authors contributed equally: Noritaka Nishida, Yuta Komori, Osamu Takarada. ✉email: shimada@iw-nmr.f.u-tokyo.ac.jp; mkikkawa@m.u-tokyo.ac.jp

Cytoplasmic dynein is a motor protein that utilizes the energy of ATP to move along microtubules (MTs) toward minus ends[1,2]. Cytoplasmic dynein is involved in many cellular processes, including spindle formation in mitosis and the intracellular transportation of various cargo molecules. As a result, mutations that affect dynein function have been implicated in various neurological diseases[3]. Within the 1.4-MDa complex that constitutes the dynein machinery[4], the heavy chain plays a central role in dynein motility. The heavy chain forms a dimer through its N-terminal tail region, which also mediates attachment to various cargo molecules. The C-terminal two-thirds of the heavy chain is called the motor domain, based on the observation that the artificially dimerized motor domain is sufficient for processive movement along MTs[5]. The motor domain has three structural elements: (1) the ATPase domain, composed of the six conserved AAA+modules (ATPase associated with diverse cellular activities, AAA1 to AAA6) that form an asymmetric ring-shaped ATPase domain; (2) the linker domain, which transmits the powerstroke movement to the N-terminal tail region; and (3) the stalk region, a 15-nm-long anti-parallel coiled-coil structure that extends from the AAA4 module and contains the microtubule-binding domain (MTBD) at the tip of the stalk. In addition, another coiled-coil structure, called the strut[6] or buttress[7], extends from the AAA5 module and forms a Y-shaped structure together with the stalk coiled-coil (Fig. 1a), and the linker also controls the conformation of the strut/buttress[8].

Based on the previous structural and functional studies[8–13], the mechanochemical cycle of the dynein motor is explained as follows[14,15]: (1) in the absence of the nucleotide, the motor domain is tightly bound to the MT. (2) ATP binding to the motor domain induces dissociation from the MT as well as the remodeling of the linker domain from straight to bent conformation (recovery stroke). (3) The motor domain searches for a new binding site on the MT, via a weak interaction mode in either the ATP or ADP/Pi state. (4) The phosphate release induces strong binding to the MT, and the linker conformation reverts from bent to straight form in the MT-bound state to transmit the powerstroke. (5) Finally, the motor domain returns to the initial state by the release of ADP from the AAA1 module.

Since the proposed model is based primarily on "snapshots" of the motor domain in the different nucleotide states in the absence of MTs, the temporal sequence of conformational changes within the large motor domain remains speculative. In particular, it is unclear how the events in the ATPase domain (ATP hydrolysis, Pi release, and power stroke) are coupled with MT binding by MTBD. The nucleotide-binding state in AAA1, especially ATP hydrolysis and the release of Pi, is critical for regulating MTBD from a low-affinity state to a high-affinity state[16,17]. On the other hand, it is known that MT binding accelerates the ATPase activity of the AAA1 site[18]. Therefore, it is unclear whether ATP hydrolysis/Pi release in the ATPase domain precedes or follows MT binding by MTBD.

Furthermore, the structural mechanism of the two-way communication between the ATPase domain and MTBD remains elusive. It has been shown that the switching of the MT affinity of MTBD is achieved by changes in the association mode of the coiled-coil, referred to as "registry". Biochemical studies have demonstrated that the sliding of CC1 with respect to CC2 by one-turn of an α−helix results in a distinctive change in the affinity for MTs[18,19]. A recent single-molecule study further demonstrated that the sliding of coiled-coil helices and the resulting changes in interaction with stalk and strut/buttress regulates the MT-binding affinity and dynein motility[8]. Comparison of crystal structures of the motor domain in the ADP/Vi-[11] and ADP[9]-bound states have indicated that the change in the registry is caused by an altered interaction between the stalk and

strut/buttress. However, the atomic detail of the affinity switching, especially regarding how the helix sliding near the ATPase domain is transmitted to MTBD, remains elusive. Currently available crystal or NMR structures that include MTBD moiety[20–22] mostly represent the low-affinity conformation, even for the full-length motor domain of *Dictyostelium discoideum* cytoplasmic dynein in the ADP-bound state[9], which would reflect the high-affinity state for MTs based on the biochemical observation. The cryo-EM-based atomic model of MT-bound MTBD in the high-affinity binding state[23,24] was used to infer that the conformational changes of the N-terminal H1 helix are essential for high-affinity binding. However, the detailed structure of the high-affinity state remains poorly defined due to the ~10 Å resolution of the cryo-EM map.

Additionally, it is unknown why the movement of the dynein motor along MTs is biased toward the minus-end direction. In contrast to kinesin and myosin motors, the two motor domains of cytoplasmic dynein are less coordinated; one motor domain is allowed to move either forward or backward with various step sizes, but the overall movement of dynein is stochastically biased in the forward direction[5]. To understand the directional preference in dynein movement, it is necessary to elucidate the binding mode of MTBD when dynein searches for a new binding site with the low-affinity state, as well as when dynein forms a final complex with the high-affinity state.

In the present study, we exploit disulfide cross-linking to regulate MTBD either in the high- or low-affinity state. We determine the NMR structures of MTBD in the low- and high-affinity states, and obtain high-resolution cryo-EM structures of MT-bound MTBD with or without pre-stabilization in the high-affinity state by the disulfide bond. Furthermore, the difference in the MT interaction sites between MTBD in the low- and high-affinity states is revealed by NMR. Those data clarified the structural changes of MTBD, explaining the affinity regulation and the two-way communication mechanism of cytoplasmic dynein, and suggested a structural mechanism for the directional preference of dynein on MT tracks.

## Results

**Construct design and characterization of MTBDs**. We generated a construct encoding a 137-residue fragment of the yeast cytoplasmic dynein; the corresponding protein contains the entire microtubule-binding domain (MTBD) and the part of the coiled-coil (Fig. 1a), as shown in a previous NMR study of mouse cytoplasmic dynein[21,25]. The protein (termed MTBD-WT) encoded by this construct was expressed in *E. coli* and purified to homogeneity.

To stabilize the conformation of MTBD in the specific MT-binding state, we introduced a disulfide bond between CC1 and CC2 to lock the registries. To stabilize MTBD in the low-affinity state, the construct was mutated such that the codons encoding S3097 in CC1 and V3222 in CC2 were mutated to instead encoded cysteines. The corresponding protein residues are in close proximity (distance between Cβ atoms: 5.7 Å) according to the crystal structure of mouse MTBD in the low-affinity state[20] (Fig. 1a, b). To stabilize MTBD in the high-affinity state, the codon for I3101 (instead of S3097) was mutated to instead encode cysteine, in combination with the V3222C mutation in CC2. The targeting of I1301, which is located one α-helical turn downstream of S3097, was based on previous biochemical studies[18] (Fig. 1a, b). The intramolecular disulfide bond formation in each of the two mutant proteins was confirmed by mobility shifts of the proteins when separated by SDS-PAGE analysis (Supplementary Fig. 1a) and by the appearance of a single set of peaks in the HSQC (heteronuclear single quantum coherence) spectra (Fig. 1f).

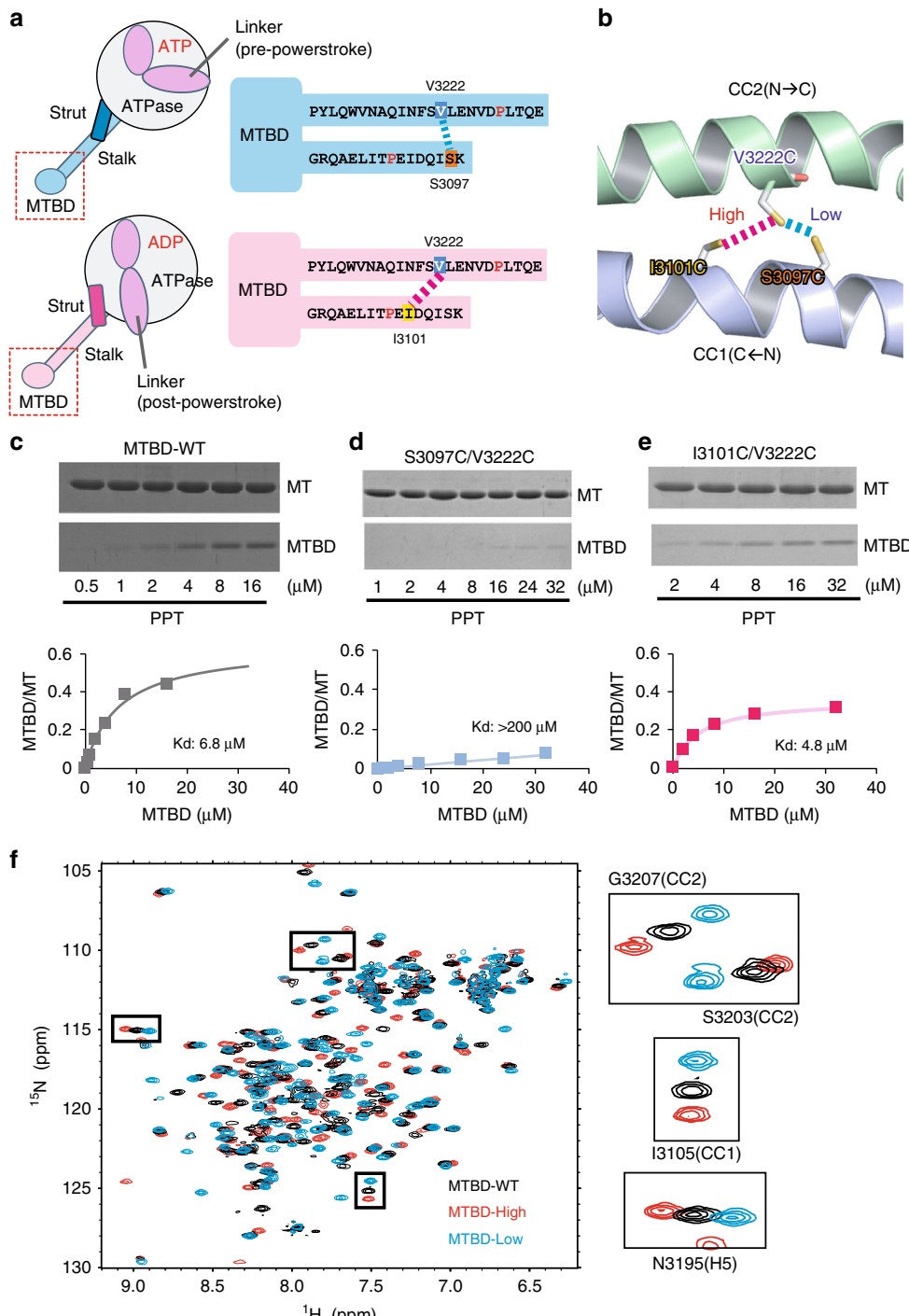

**Fig. 1 Design of MTBD constructs containing a disulfide bond within the coiled-coil. a, b** Schematic drawings of the MTBD mutants stabilized either in high-affinity (pink) or low-affinity (cyan) states. The positions of the residues mutated to cysteine (S3097C, I3101C, V3222C) are shown schematically (**a**) or in the homology model of yeast MTBD using the mouse SRS (Seryl-tRNA synthetase) chimera (PDB code 3ERR) as a template (**b**). **c–e** The analysis of MT-binding affinity of (**c**) MTBD-WT, (**d**) MTBD-Low, and (**e**) MTBD-High by co-sedimentation assay. Source data are provided as a Source Data File. **f** The overlaid $^1$H $^{15}$N HSQC (heteronuclear single quantum) spectra of MTBD-WT (black), MTBD-Low (cyan), and MTBD-High (pink). Selected signals are enlarged in the boxes to the right side.

To examine whether those mutant proteins indeed are stabilized in low- and high-affinity states, we performed co-sedimentation assays. While the dissociation constant (Kd) of MTBD-WT was estimated to be 6.7 μM (Fig. 1c), MTBD (S3097C/V3222C) exhibited significantly lower MT affinity (Kd > 200 μM) (Fig. 1d), indicating that the disulfide bond formation stabilized MTBD conformation in a low-affinity state. On the other hand,

the Kd of MTBD (I3101C/V3222C) was 4.8 μM (Fig. 1e), a value that was much smaller than that of the S3097C/V322C mutant and slightly smaller than that of MTBD-WT. This result indicated that the I3101C/V3222C disulfide bond locked MTBD in a stronger-binding state. Hereafter, we refer to MTBD(S3097C/V3222C) and MTBD(I3101C/V3222C) mutants as MTBD-Low and MTBD-High, respectively.

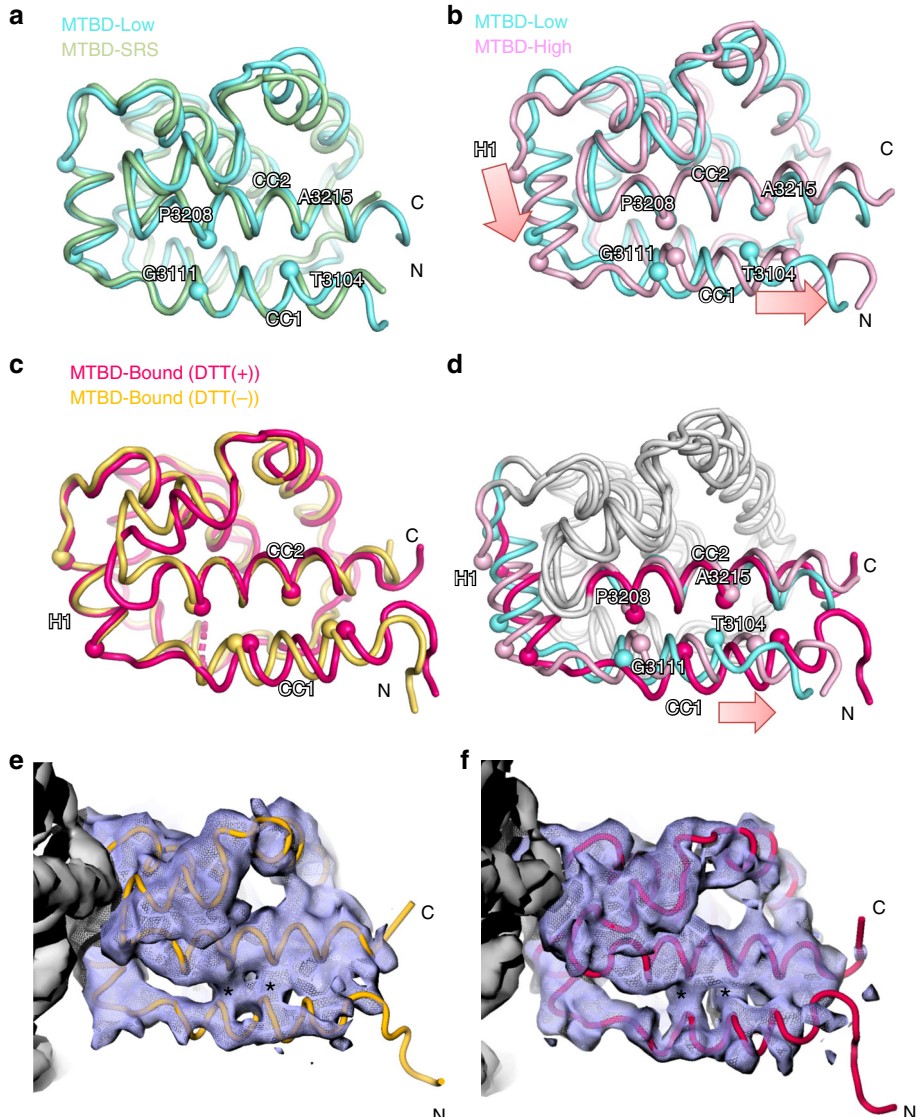

**Fig. 2 Structure comparison of MTBD-High, MTBD-Low and MTBD in complex with MT. a** Superposition of MTBD-Low (cyan) and mouse MTBD SRS chimera (PDB code 3EER, light green) viewed from the side. **b** Superposition of MTBD-High (pink) and MTBD-Low (cyan). **c** Superposition of cryo-EM model of MTBD in complex with MTs in the DTT(−) (orange) and DTT(+) (magenta) conditions. **d** The superposition of MTBD-Low (cyan), MTBD-High (pink), and MTBD-Bound (magenta). Only the CC1, H1, and CC2 moieties are colored. The Cα positions of CC1 and CC2 residues that signify the coiled-coil registry are shown by spheres. The notable conformational changes of H1 and CC1 are indicated by pink arrows. **e**, **f** Superposition of the cryo-EM map filtered to 4 Å and the model of MTBD in complex with MTs in (**e**) DTT(−) (yellow) and in (**f**) DTT(+) (magenta) conditions. Asterisks (*) are placed on the regions corresponding to the hydrophobic interactions between CC1 and CC2 in both maps.

**Solution NMR structures of MTBDs**. To gain structural insights into the affinity switching mechanism of MTBD, we derived the solution structures of MTBD-High and MTBD-Low using the standard NMR-derived distance restraints (Supplementary Fig. 1b, c, Supplementary Table 1). Both MTBD-High and MTBD-Low were composed of six α-helices (H1 to H6) and an anti-parallel coiled-coil (CC1 and CC2, Fig. 2). The coiled-coil registry of CC1-CC2 in MTBD-Low was similar to the registry of mouse MTBD in the low-affinity states[20], corresponding to the +β-registry (Fig. 2a). On the other hand, MTBD-High showed a distinctive slide of the CC1 helix with respect to CC2 by an α-helical half-turn, accompanied by the downward shift of H1 with respect to H3 (Fig. 2b). We refer to the registry of MTBD-High as "semi-α", since the sliding of CC1 places that helix in an intermediate position between the +β- and α-registries. We compared the overall structures of MTBD-High and MTBD-Low with the existing MTBD atomic models in the low-affinity states[9,11,20,22]. Although the construct design and the source species of dynein were different, all previous structures showed H1 and CC1 in positions similar to those seen in MTBD-Low (Supplementary Fig. 1d and e), indicating that the conformational changes of H1 and CC1 from MTBD-Low to MTBD-High are related to the increase in MT affinity. In addition, the HSQC spectrum of MTBD-WT indicated that signals originating from the residues in CC1 and CC2 were located between MTBD-High and MTBD-Low (Fig. 1f), implying that MTBD without the disulfide bond is in an equilibrium between the +β and semi-α registries, and suggesting that the population shift toward the semi-α registry correlates with the increase in MT-binding affinity.

**Cryo-EM structure and modeling of the MTBD-MT complex.** To elucidate the structural basis of the MT binding of MTBD, we

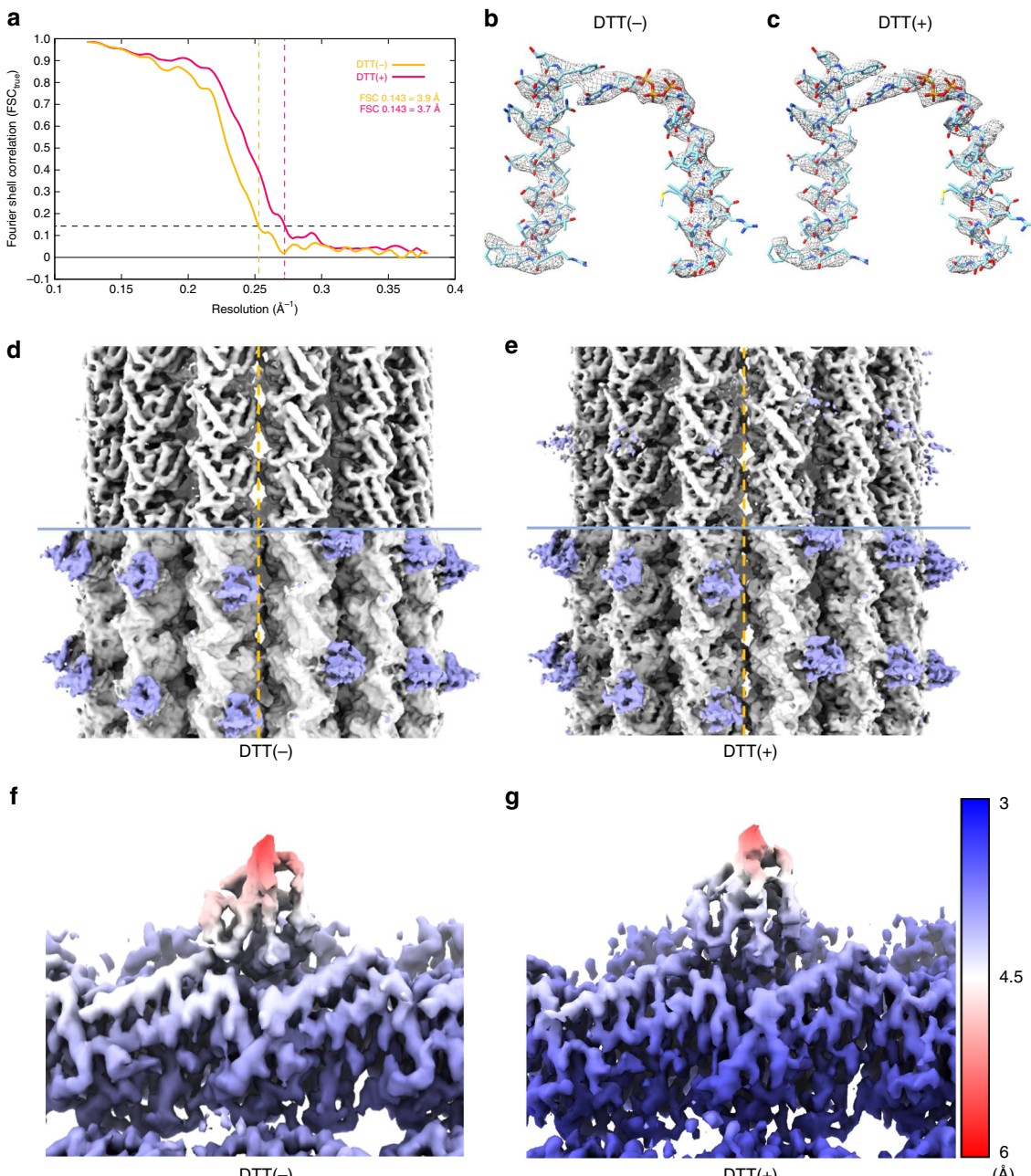

**Fig. 3 Cryo-EM single-particle reconstruction of the MTBD-High-MT complex in the absence (−) and presence (+) of DTT. a** Fourier Shell Correlation (FSC$_{true}$) curves for the structures of MTBD-High-MT complex for both ± DTT conditions, using a FSC 0.143 criterion. **b, c** Cryo-EM density segmented from the post-processed maps (using *relion_postprocess*, filtered to local resolution) of MTBD-High-MT complex and fitted atomic model of H4, H7, GMPCPP, and Mg$^{2+}$ of α-tubulin (PDB code: 1JFF) for (**b**) DTT(−) and (**c**) DTT(+) conditions. **d, e** C1 reconstruction of MTBD-High-MT complex for (**d**) DTT(−) and (**e**) DTT(+) conditions (filtered to 4.44 Å and 4.33 Å, respectively). The broken line indicates the location of the seam. The upper half is displayed at high threshold to show secondary structures in MTs. The lower half is displayed at lower threshold to show the MTBD structure. **f, g** Density of MTBD-High-MT complex colored and filtered according to local resolution determined using *relion_postprocess* for (**f**) DTT(−) and (**g**) DTT(+) conditions.

performed cryo-EM single-particle reconstruction of MTBD-High in complex with MTs (see Methods and Supplementary Fig. 2). We also analyzed the complex in the presence of DTT (dithiothreitol) to examine whether the disulfide bond formation in the coiled-coil affected the MTBD structure in the MT-bound state. Without imposing pseudo-helical symmetry on MT, the structures (including the seam) were well resolved, with a density corresponding to MTBD seen at the intradimer boundary of αβ-tubulin (Fig. 3b-g). Using a cut-off of 0.143 for the Fourier Shell

Correlation (FSC$_{true}$, see Methods for detail), the overall resolution for each map was estimated to be 3.9 Å and 3.7 Å in the absence (−) and presence (+) of DTT (hereafter referred as ±DTT), respectively (Fig. 3a, Supplementary Table 2).

Since the local resolution of the cryo-EM density of MTBD is ~4–5 Å (Fig. 3f, g), which is not sufficient for constructing the de novo atomic model, we built an atomic model structure of the MTBD-High-MT complexes by a flexible-fitting method. The atomic models of tubulin dimer (PDB code: 1JFF) and

NMR-derived MTBD-High were fit into the map, and subjected to flexible fitting using MDFF and energy minimization[26] (Supplementary Fig. 3a, b). In both the +/− DTT complex structures, all helices of MTBD were well accommodated within the maps (Supplementary Fig. 3c, d). Three trials of the flexible docking resulted in similar MTBD atomic models; we selected the model with the highest cross-correlation coefficient. Although the density corresponding to the CC1 region was relatively weak, the CC1 helix was well fit into the map for both DTT conditions (Fig. 2e, f, Supplementary Fig. 3c, d). Interestingly, we found that the DTT(−) and DTT(+) structures were very similar to each other (Fig. 2c), suggesting that MT binding, not the disulfide bond, determines the MT-bound MTBD-High structure. In addition, the good agreement of cryo-EM density corresponding to the hydrophobic interactions between CC1 and CC2 (Fig. 2e, f, asterisks) in both conditions suggests the similarity of the two structures and the validity of fitting. Therefore, in the following, we mainly used the atomic model of the MTBD-High in complex with MT in the presence of DTT, which we refer to as MTBD-Bound.

To examine the structural changes of MTBD caused by MT binding, we compared the structures of the MTBD-High, MTBD-Low, and the MTBD-Bound models (Fig. 2d). The MTBD-Bound model exhibited prominent conformational changes at the CC1 and H1 region, compared to MTBD-Low and MTBD-High. Notably, CC1 of MTBD-Bound was positioned further toward the N-terminus compared to MTBD-High (corresponding to semi-α-registry), resulting in a one-turn α-helical shift in registry from MTBD-Low (corresponding to +β-registry), i.e. converting the structure into an α-registry (Fig. 2d). Therefore, MTBD in the α-registry is likely stabilized upon MT binding, presumably by an induced-fit mechanism.

**Residues involved in interactions between MTs and MTBD.** Next, we investigated the residues involved in high-affinity binding in the MTBD-MT complex. Overall, the MT-binding surface of MTBD is rich in positively charged residues, while the MTBD-binding surface of MTs is occupied predominantly by negatively charged residues. MTBD binds to α- and β-tubulin via the H1, H3, and H6 helices (Fig. 4a), forming six pairs of salt bridges. On the α-tubulin side, R402(α) forms a salt bridge with H1's E3122, the sole negatively charged residue of the MT-binding surface on MTBD (Fig. 4b). E415(α), which forms an intramolecular salt bridge with R402(α) in the original tubulin dimer structure, engages in an intermolecular salt bridge with R3201 on H6, together with E420(α) (Fig. 4b). These charge-flipped interactions also were observed in the cryo-EM model of MTBD of *Dictyostelium* dynein[24]. On the β-tubulin side, D427(β) and E199(β) form a salt bridge with K3117 and R3124 on H1, respectively (Fig. 4c). In addition, E159(β) and D163(β) form a salt bridge with R3159 and R3152 on H3, respectively (Fig. 4d). Therefore, our results suggested that the high-affinity state of MTBD-High is achieved by formation of multiple salt bridges between MTBD and MTs on the complementarily shaped surface.

To validate our high-affinity model, we performed mutational analysis of the positively residues hypothesized to be involved in the salt bridge in the complex to non-charged residues (either Ala or Met). Indeed, mutation of each of these residues resulted in greater than 2-fold decrease in MT affinity (Supplementary Fig. 4). Notably, mutation of K3116 or of K3204, both of which are located near the MT-binding site but are not thought to be directly involved in the interactions, yielded MTBD-High proteins with MT affinities similar to that of the original MTBD-High protein.

**Structural basis of affinity switching.** Next, we explored the affinity switching mechanism between MTBD in the low-affinity

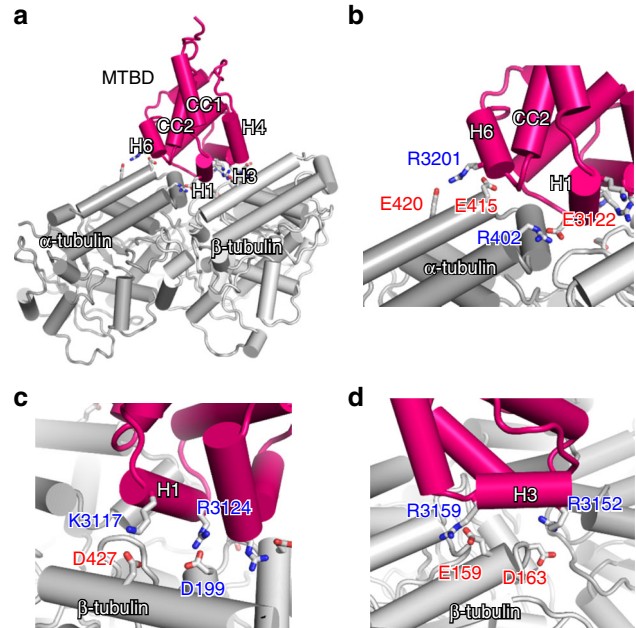

**Fig. 4 The interaction mode of the MTBD and tubulin dimer. a** Overall structure of the docking model obtained by MDFF for MTBD, α-tubulin, and β-tubulin (colored magenta, dark gray, and light gray, respectively). **b–d** The close-up view of the interface of the (**b**) α-tubulin side, (**c**) H1 helix, and (**d**) H3 helix. The residues involved in intermolecular salt bridges are shown by stick models.

and high-affinity states using NMR. Since cryo-EM analysis of MTBD-Low in complex with MTs was unsuccessful (presumably due to MTBD-Low's weak affinity for MTs), we investigated the difference in the MT-binding surface of MTBD-Low and MTBD-High using the transferred cross-saturation (TCS) method[27]. The TCS method allowed us to identify the MT-contacting surface of MTBD by observing the saturation transfer from MT to MTBD as intensity reductions in the NMR signals. In TCS experiments using MTBD-High, the residues in H1, H3, and the loop near H6 exhibited greater signal intensity reductions compared to other residues (Fig. 5a, c). Note that all three of these helices also were located at the interface between MTBD and MT in our cryo-EM-based models (Fig. 5d). On the other hand, the TCS experiments with MTBD-Low showed greater intensity reduction for only one residue in H3, while the signals from the similar residues in H1 and the loop near H6 were reduced (Fig. 5b, e). These results indicated that H3 makes a crucial contribution in the affinity switching of MTBD.

**Discussion**

In the present study, we used a combination of structural methods to demonstrate that dynein MTBD can assume three major states, namely the +β, semi-α, and α-registries (Fig. 6a and Supplementary Movie 1), which may explain the affinity switching mechanism of MTBD. In MTBD-Low, MTBD adopts a +β-registry, with the H1 and H6 regions forming a weak-binding surface for MTs (Fig. 6a, left); in MTBD-High, MTBD adopts a semi-α-registry with sliding of CC1 by a half-turn of the α-helix, with the H1, H3, and H6 regions forming a strong-binding surface (Fig. 6a, middle). In the MT-bound state, MTBD undergoes a further conformational change to assume the α-registry by sliding of one-turn of an α-helix in total (Fig. 6a, right).

To see whether the structural changes of the MTBD-MT complex are shared among species, we compared our model

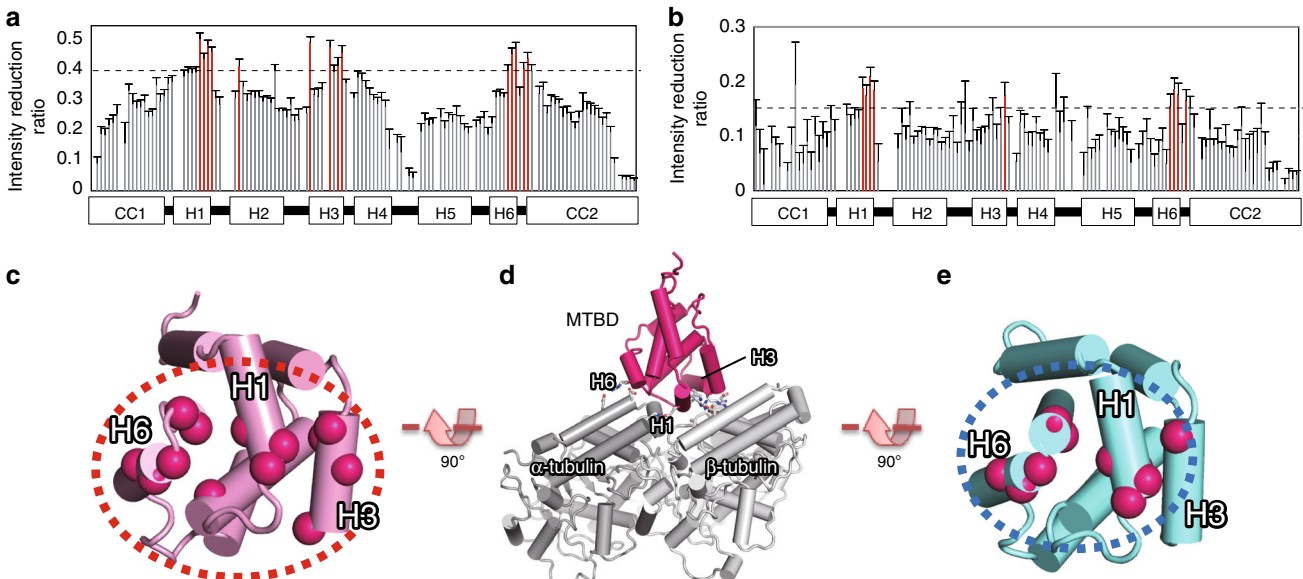

**Fig. 5 Identification of the low- and high-affinity interface by TCS (transferred cross-saturation) experiments. a, b** Plots of the intensity reduction ratio of the amide resonances of (**a**) MTBD-High and (**b**) MTBD-Low in TCS experiments. The error bars are calculated based on signal-to-noise ratios. Residues showing an intensity reduction ratio greater than 0.15 for MTBD-Low or greater than 0.4 for MTBD-High are colored red. Signals with standard deviations greater than 0.1 were excluded. Although V3133 in H2, which is not exposed to the surface of MTBD, showed a significant intensity reduction in TCS experiments of MTBD-High, this reduction was presumably due to spin diffusion from H3 residues. Source data are provided as a Source Data File. **c–e** Mapping of the residues with significant intensity reduction in TCS experiments of (**c**) MTBD-High and (**e**) MTBD-Low. The overall orientation of MTBD with respect to MTs is shown in (**d**).

with those proposed in previous reports (Supplementary Fig. 5a–c)[23,24,28]. Among the existing studies, a recent report by Lacey et al.[28] showed MTBD structures of mouse cytoplasmic dynein and human axonemal dynein resolved to resolutions (4.1 and 4.5 Å, respectively) similar to ours. These two structures showed better agreement with our model than with other models proposed by Uchimura et al.[24] and Redwine et al.[23] (Supplementary Fig. 5e). Therefore, we compare our structures primarily with those proposed by Lacey et al.

In addition, the two structures provided by Lacey et al. exhibited significant displacement of H1 and sliding of the CC1 helix, corresponding to a one-turn slide of the CC1 α-helix with respect to CC2 (Supplementary Fig. 5d), a point that was not clearly mentioned in Lacey et al. due to the lower resolution of the MTBD moiety (5–7 Å) in their work compared to our structure (4–5 Å). Although their MTBD structure is pre-stabilized in the high-affinity state using an SRS (Seryl-tRNA synthetase)-chimera construct, our results demonstrated that the conformations of MTBD in complex with MT are similar regardless of high-affinity restraints (i.e., with or without DTT treatment), indicating that MT binding induces the sliding of CC1. As revealed by our TCS experiments, H3 plays a significant role in the high-affinity binding, contributing to the formation of two salt bridges (between R3152 and E159(β), and between R3159 and D163(β)) (Fig. 5c). In the Lacey model, two conserved positively charged residues, R3334 (equivalent to R3152) and R3342 (equivalent to R3159), are positioned close enough to form salt bridges with negatively charged residues of β-tubulin (Supplementary Fig. 6a, b). On the other hand, these positively charged H3 residues are not conserved in MTBD of the *Dictyostelium* dynein (Supplementary Fig. 6e). Instead, K3424 and R3423, which are positioned in the middle of the H3 helix, could serve as alternative salt bridge partners. Indeed, K3424 formed a salt bridge with E159(β) (Supplementary Fig. 6c), and reorienting of the side chain of R3423 enables formation of a salt bridge with D163(β) (Supplementary Fig. 6d). Therefore, it appears that the

two salt bridges formed on H3 are also universally conserved, participating in high-affinity binding by dynein molecules of multiple species.

Based on the modeled conformational changes of MTBD in the absence and presence of MTs in the present study, we propose that the mechanochemical cycle of dynein proceeds via two distinctive pathways, namely an "ATPase-driven pathway" and a "MT-binding-induced pathway". In the ATP-bound states, the stalk coiled-coil adopts the +β-registry across the entire region following the linker detachment from AAA5 (Fig.6b (2)), and MTBD is stabilized in the low-affinity state[8] (Fig. 6b (3)). In the ATPase-driven pathway (Fig. 6b; shown in light green), ATP hydrolysis (Fig. 6b (4–1)) and subsequent release of Pi induces conformational changes of the ATPase domain, which alters the interaction mode between the stalk and strut coiled-coils to induce the release of the stalk coiled-coil association mode from the +β-registry[9,29]. The perturbation of coiled-coil packing induces conformational equilibrium between low-affinity (+β) and high-affinity (semi-α) states, thereby increasing the MT-binding affinity of MTBD as MTBD searches for a new MT binding site (Fig. 6b (4–2)). The complex formation with MTs stabilizes the conformation of coiled coil in the α-registry near the MTBD region, a change that is propagated through the entire coiled-coil helix toward the ATPase domain (Fig. 6b (4–3)). Consequently, further conformational rearrangements in the ATPase domain induce the powerstroke of the linker (docked to AAA5) and a shift of the entire stalk in the α-registry (Fig. 6b (5)). The release of ADP returns the dynein to the original state of the mechanochemical cycle (Fig. 6b (1)). In the ATPase driven-model, ATP hydrolysis occurs prior to MT-binding, thus cannot explain why ATPase activity is enhanced in the presence of MTs. As an alternative model, we considered the MT-binding-induced pathway. For the dynein motor in the weakly binding β-registry (Fig.6 (2)), MT-binding can induce the conformational changes of MTBD and the coiled-coil registry is shifted directly from +β to α-registry at the MTBD

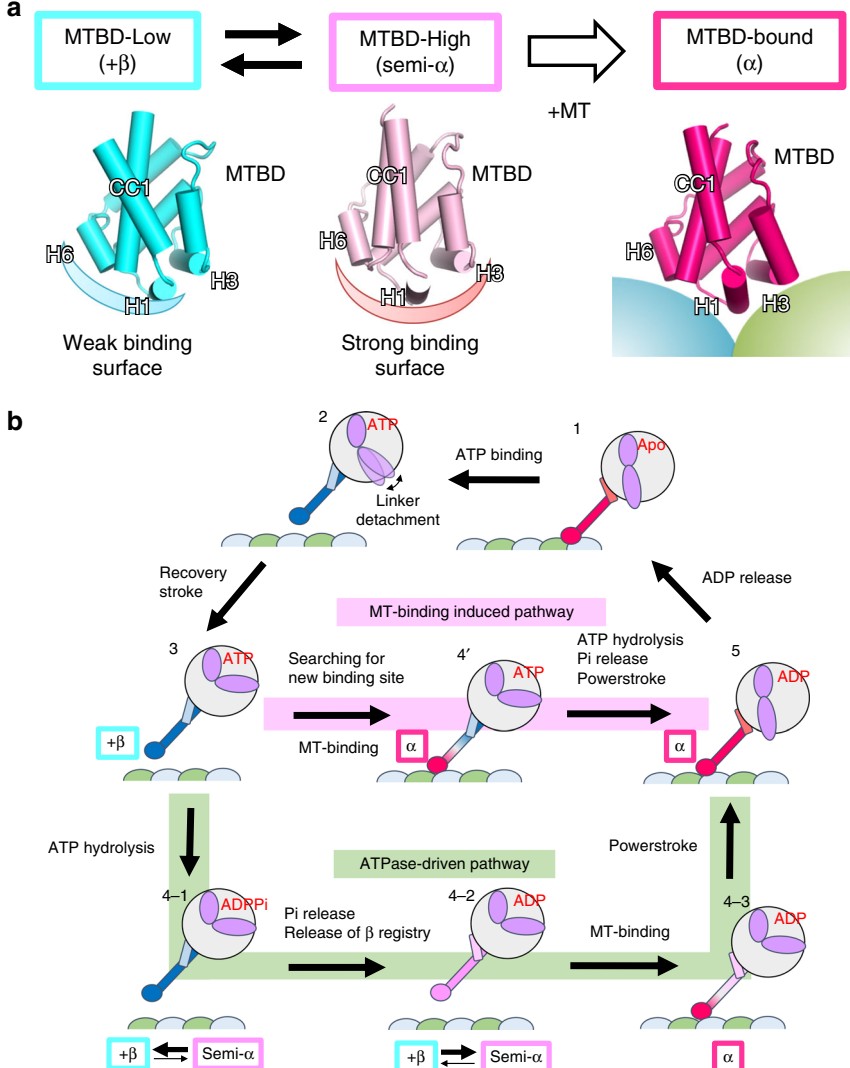

**Fig. 6 The proposed mechanism for two-way communications between MTBD and ATPase domain during the mechanochemical cycle of dynein. a** The three conformational states of MTBD revealed in this study. In the MT-unbound state, MTBD undergoes conformational equilibrium between MTBD-Low (left, +β-registry) and MTBD-High (middle, semi-α-registry). MT binding induces conformational changes of MTBD with sliding of CC1 (α-registry). **b** Mechanochemical cycles of dynein. In the ATPase-driven pathway (green), the hydrolysis of ATP (3 to 4–1) and subsequent release of Pi (4–1 to 4–2) shift the equilibrium of MTBD toward semi-α state (4–2) to increase the MT-binding affinity. After MT binding, the stabilized α-registry transmits the conformational changes to ATPase domain, thereby facilitating the powerstroke movements of the linker (4–3 to 5). In the MT-binding-induced pathway (pink), the ATP-bound dynein searches for a new binding site in the low-affinity state. Upon MT binding, the stabilization of the α-registry transmits the conformational changes to the ATPase domain, which facilitates ATP hydrolysis, Pi release, and the powerstroke all at once.

proximal region (Fig. 6b (4')). As a result, the conformational changes are transmitted toward the ATPase domain to facilitate ATP hydrolysis, release of Pi, and the powerstroke of the linker, all as a consequence of MT binding (Fig. 6b; highlighted in pink). Therefore, we propose that the three conformational states of MTBD mediate the two-way communication between MTBD and the ATPase domain via either ATP-hydrolysis-dependent or MT-binding-driven pathways.

A recent single-molecule study demonstrated that the introduction of a disulfide bond at a position midway between the α and β-registries stabilizes the dynein motor in a distinctive functional state, referred to as γ-registry[8]. Characterization by optical tweezer experiments showed that γ-registry dynein possesses a force resistance profile similar to that of WT dynein when the force was applied in the forward direction (i.e. corresponding to the movement of the trailing head). The γ-registry is predicted to causes sliding of the stalk α-helix by a half-turn, similar to the

semi-α-registry in our study. However, there are several differences between the semi-α and γ-registries. First, the semi-α conformation was observed only in the absence of MTs; MT-bound MTBD assumed the α-registry in the present study. In contrast, the γ-registry was observed in the MT-bound state whether the force was applied from either the forward or backward direction. Second, the position of disulfide cross-linking also differed between the two studies (the disulfide cross-linking was close to MTBD with semi-α, but in the middle of sthe talk with γ-registry). Since the γ-registry was characterized primarily by single-molecule assays, further structural studies of the dynein stalk possessing a disulfide bond at the γ-registry position will be required to determine whether the γ-registry represents a structure similar to the semi-α-registry.

Our result also may provide a clue to the mechanism of the biased movement of dynein along MTs. It is known that the movement of the dynein motor along MTs is biased in the

**Fig. 7 A model for the directional bias for the applied forces and movement direction of dynein. a** MTBD of dynein makes an initial attachment via a weak-binding interface (H1 and H6). **b** Backward force or diffusive search in the forward direction tilts MTBD, rendering H3 proximate to the binding site of β-tubulin (indicated by an arrow). **c** MT binding by MTBD via a strong-binding interface (H1, H3, and H6) induces the sliding of CC1, which would transmit the signal toward the ATPase domain to facilitate ATP hydrolysis and the release of Pi.

forward (i.e., minus-end) direction[5], and dyneins on MTs exhibit stronger resistance to backward (i.e., plus-end) force than to forward force[30]. We propose that the force-direction-dependent binding of dynein can be explained as follows: in searching for a new docking site, MTBD initially binds to MTs in the weak-binding β-registry conformation[8] via H1 and H6, while the interaction mediated by H3 has not formed yet (Fig. 7a). When force is applied in the backward direction, the stalk would tilt backward. If the stalk is stiff enough to transmit the force, H3 moves toward β-tubulin (Fig. 7b), which would be preferable for high-affinity binding (Fig. 7c). In two-headed motor domain stepping, the leading head of dimeric dynein in the ATP-bound (low-affinity) state searches for a new binding site, while the lagging head in the ADP or apo (high-affinity) state remains attached to the MT. The angle between the stalk of the leading head and the MT would become smaller when the leading head steps forward, than when that head steps backward. Therefore, for the same reason as when the backward force is applied, the forward movement would be favored over the backward. EM studies have indicated that the naïve dynein stalk in the MT-bound state tilts backward, with an angle of ~42°[31]. A recent study showed that the reverse kink mutant, which moves in the opposite direction, still possesses resistance to the minus-end force (i.e., as with the regular dynein motor)[32], further supporting the binding mode of low-affinity MTBD and confirming that its interface geometry with respect to MTs is crucial for generating the directional preference of dynein motility.

## Methods

**Protein preparation and characterization.** The MTBD construct encoding residues K3096-E3232 of the cytoplasmic dynein of *Saccharomyces cerevisiae* was expressed and the resulting protein purified according to the previous report[33]. Briefly, the pET-15b plasmids (Merk Millipore) harboring MTBD-WT, MTBD-High (I3101C/V3222C) and MTBD-Low (S3097C/V3222C) –encoding sequences were transformed into *E. coli* BL21 (DE3) codon plus RP strain (Agilent Technologies, Catalog Code:230255). The resulting cells were cultured in M9 minimal medium and the proteins were induced with 0.4 mM IPTG at 20 °C for 6 h. The harvested cells were lysed by sonication, and the supernatant was applied to HIS-select Nickel affinity chromatography (Sigma); bound proteins were eluted with imidazole. Subsequently, the N-terminal His-tag was cleaved by thrombin (Novagen) treatment, and the MTBD proteins were further purified by size exclusion chromatography using a Superdex$^{TM}$ 75 pg column (GE Healthcare). Sequences of the primers used to construct the plasmids are provided in Supplementary Table 3.

**Co-sedimentation assay.** Porcine tubulin was polymerized in PEMG buffer (100 mM PIPES, pH 6.8, 1 mM EGTA, 1 mM MgCl$_2$, 1 mM GTP) supplemented with 5% DMSO for 30 min at 37 °C, and stabilized by addition of 10 μM paclitaxel. Purified MTBDs (at the concentrations ranging from 1-32 μM) were mixed with 5 μM porcine microtubules (MTs) in the binding buffer (100 mM PIPES, pH 6.8, 50 mM NaCl, 2 mM MgCl$_2$, 1 mM EGTA), and the mixtures were centrifuged at 210,000 × *g*, 25 °C, 10 min, using an Optima MAX equipped with an MLA80 rotor (Beckman Coulter). The resulting pellets were rinsed once with 50 μL of the binding buffer, and then subjected to SDS-PAGE. The bands were analyzed by ImageJ software (http://rsb.info.nih.gov/nih-image/) for densitometric analysis. The dissociation constants (Kd) were obtained by one-site curve-fitting analysis

using Origin 5.0 (Microcal Software). Uncropped images of these SDS-PAGE gel are provided in the Source Data file.

**NMR analysis and structure determination.** All NMR experiments were performed using an AVANCE 800 or AVANCE 500 spectrometer equipped with a TCI cryogenic probe (Bruker Biospin) at 25 °C. The backbone and side-chain assignments of MTBD-High and MTBD-Low were performed using a series of triple-resonance NMR experiments[33]. Inter-proton distance information was obtained from the $^{13}$C-edited NOESY-HSQC and $^{15}$N-edited NOESY-HSQC spectra. The mixing time was 100 ms for all NOESY experiments. A series of $^{1}$H-$^{15}$N HSQC spectra was acquired on a sample freshly dissolved in D$_2$O, to obtain a backbone hydrogen-bonding amide group. In addition, dihedral angle constraints based on HNHA[34] and TALOS+[35] were used. The $^{1}$H-$^{15}$N residual dipolar coupling constants were collected using $^{15}$N-labeled MTBD in the presence of 8 mg/mL of pf1 phage and 4.2% C$_{12}$E$_5$/n-hexanol bicelle (0.96:0.04) using IPAP (in-phase anti-phase) HSQC experiments[36]. The structure calculation was initially performed by Cyana2.1 using the above distance constraints without RDC information[37]. The final structure ensemble was calculated by Xplor-NIH[38], including the RDC constraints and Cyana-assisted NOE assignments; the distance constraints are summarized in Supplementary Table 1. The 10-lowest-energy structures were selected as a final ensemble. The percentages of residues in the favored, allowed and outlier regions of the Ramachandran map were 87.9, 9.8, and 2.3, respectively, for MTBD-Low, and 82.7, 11.6, and 6.9, respectively, for MTBD-High, as determined by the MolProbity web server (http://molprobity.biochem.duke.edu/).

**Transferred cross-saturation (TCS) experiments.** $^2$H-, $^{15}$N-labeled MTBD was mixed with polymerized MT at a 10:1 ratio in the NMR buffer (10 mM NaPi, pH 7.0, 200 mM NaCl, 80% D$_2$O). The TCS experiments were performed using an AVANCE 800 NMR spectrometer equipped with a cryogenic TCI probe, employing the previously reported pulse program[27]. A pair of HSQC spectra was collected with and without radio frequency irradiation at 1.00 ppm for 0.5 s at the end of the recycling delay period (total duration, 4.0 s) using the I-BURP2 pulse scheme (maximum B1 amplitude 0.5 kHz). The intensity ratios of each peak in the two HSQC spectra were analyzed by SPARKY software (UCSF).

**Cryo-EM sample preparation of MTBD-High-MT complex.** For cryo-EM observation, GMPCPP-MT was polymerized by incubating 0.18 mg/mL GMPCPP-tubulin in PEM/NP40 buffer (80 mM PIPES pH 6.8, 1 mM EGTA, 1 mM MgCl$_2$, 0.01% NP40) for 30 min at 37 °C. For MTBD-High-MT complex, 3 μL of the GMPCPP-MT solution was applied to a glow-discharged, carbon-coated grid (Quantifoil R 1.2/1.3) inside a Vitrobot (Thermo Fisher Scientific) and incubated for 30 s at 22 °C and 100% humidity. The grid was washed once with 3 μL of 1.1 mg/mL MTBD solution and once with 3 μL of PEM/NP40 buffer (30-s incubation each), followed by blotting and vitrifying in liquid ethane. To observe the reduced form of MTBD in complex with MTs, GMPCPP-MT was polymerized by incubating 1.0 mg/mL GMPCPP-tubulin in PEM buffer (80 mM PIPES pH 6.8, 1 mM EGTA and 1 mM MgCl$_2$) for 30 min at 37 °C. MTBD-High, NP40 and DTT were added to the GMPCPP-MT solution, to the final concentration of 0.8 mg/mL MTBD-High, 0.4 mg/mL GMPCPP-MT, 0.01% NP40 and 1 mM DTT, and the mixtures were maintained for 2 h at room temperature. Disulfide bond reduction of MTBD-High was confirmed by SDS-PAGE analysis (Supplementary Fig. 1f). An aliquot (2 μL) of the specimen was applied to a glow-discharged, carbon-coated grid (Quantifoil R 1.2/1.3) inside a Vitrobot set at 6 °C and 100% humidity and incubated for 10 s before blotting and vitrifying in liquid ethane.

**Cryo-EM data collection for MTBD-High-MT complex.** A large cryo-EM dataset of the MTBD-High-decorated GMPCPP-MT in the absence (−) or presence (+) of DTT was collected on a 200-KeV Talos Arctica electron microscope (Thermo Fisher Scientific). A Gatan Image Filter (GIF) was used for data collection, with a slit width of 30 eV. Defocus ranging from −1.0 to −2.5 μm was used. Totals of 1820 and 621 movie stacks for the DTT(−) and DTT(+) conditions, respectively,

were recorded on a post-GIF K2 Summit camera (Gatan) in counting mode with a calibrated pixel size of 1.32 Å per physical pixel and a dose-rate of ~10 electrons/pixel/s. Each exposure was 10 s long and recorded as a movie of 40 frames, corresponding to a dose of 1.35 e/Å$^2$ for each frame, and a cumulative dose of 54.0 e/Å$^2$ on the specimen. The data were collected semi-automatically using SerialEM[39].

**Image processing**. For the collected data, drift correction was performed using MotionCor2[40]. The contrast transfer function parameters were estimated from the motion-corrected micrographs using Gctf[41]. In the next step, we used PyFilamentPicker (manuscript in preparation) to semi-automatically select 14-PF MTs and to cut each MT into overlapping boxes with an 82-Å non-overlapping region along the MT axis between adjacent boxes (Supplementary Table 2). We used a "super-particle"-based approach[42] to determine the correct seam location for each particle. The seam-finding process was repeated several times to ensure accurate seam determination (Supplementary Fig. 2). During the iteration, "bad" MTs (that is, those whose seam location was not determined consistently) were removed (Supplementary Table 2). After several cycles, we calculated C1 reconstruction and confirmed whether the seam could be clearly seen, meaning seam locations for the majority of particles were correctly determined (Fig. 3d, e). We used *relion_postprocess* in RELION-3 package1 for estimating final resolution and B-factor for map sharpening. The resolution was estimated by calculating the Fourier Shell Correlation ($FSC_t$) of one "good" protofilament segment containing three central adjacent tubulins from two half-maps reconstructed from randomly separated particles using FREALIGN (Symmetry: HP)[43] (Supplementary Table 2). As a last step, we calculated $FSC_{true}$ by additionally calculating $FSC_n$, FSC for the high-resolution noise (<10 Å) -substituted stack (Fig. 3a, Supplementary Table 2)[44]. $FSC_{true}$ was evaluated as follows (Eq. 1).

$$FSC_{true} = (FSC_t - FSC_n)/(1 - FSC_n) \qquad (1)$$

The final resolution for the reconstructed maps was determined using an FSC 0.143 criterion for $FSC_{true}$. There were no significant gaps in resolution between the resolutions evaluated using $FSC_t$ and $FSC_{true}$. The local resolution was calculated and the estimated B-factor was applied for visualization using *relion_postprocess* (Fig. 3f, g).

**Molecular dynamic by flexible fitting**. The density map containing a single unit of tubulin dimer/MTBD complex was extracted from the original map using the e2proc3d.py program in EMAN2. The atomic structure of the tubulin dimer (PDB code: 1JFF) was fitted into the segmented map using UCSF chimera 1.10, and the NMR structure of MTBD-High was placed manually in the initial model. The molecular dynamics flexible fitting (MDFF) was performed following the tutorial provided by the developer using the explicit solvent mode. Briefly, the initial atomic model was solvated and ionized by VMD software for MDFF simulation in the explicit solvent. The MDFF simulation was performed using the software NAMD 2.1, for 100,000 steps using a force-scaling factor (ξ) of 1.0 kcal/mol for both ±DTT cryo-EM maps, in the presence of secondary structure restraints and restraints to prevent *cis/trans* peptide transitions and chirality errors. Following the MDFF simulation, a simple energy minimization was performed, for 10,000 steps using a force-scaling factor (ξ) of 10.0 kcal for both maps. It should be noted that the side-chain orientations of R402(α) and E415(β), which form an intramolecular salt bridge in the original tubulin dimer coordinates (1JFF), was manipulated by PyMOL so that the side-chains were exposed to the solvent to enable the formation of intermolecular salt bridges. The trajectory of the MDFF simulation was analyzed by VMD, to calculate the cross-correlation coefficient (CCC) between the target density map and each frame of the MDFF simulation. The CCC was improved from 0.60 to 0.67 for the DTT(−) structure and from 0.58 to 0.67 for the DTT(+) structure, at which point the CCC reached a plateau.

**Reporting summary**. Further information on research design is available in the Nature Research Reporting Summary linked to this article.

## Data availability

Data supporting the findings of this paper are available from the corresponding authors upon reasonable request. A reporting summary for this Article is available as a Supplementary Information file.

The source data underlying Figs. 1c–e, 3a and 5a, b, Supplementary Fig. 1a and f, and Supplementary Table 2 are provided as a Source Data file.

The NMR structure coordinates and the chemical shift data of MTBD-High and -Low have been deposited in the Protein Data Bank (PDB) and Biological Magnetic Resonance Bank (BMRB): MTBD-High (PDB-6KJN and BMRB-11490) and MTBD-Low (PDB-6KJO and BMRB-11495). The cryo-EM density maps and atomic coordinates of +/− DTT MTBD-High-MT complexes have been deposited in the Electron Microscopy Data Bank (EMDB) and PDB: MTBD-High-MT complex DTT(-) (EMD-9996, PDB-6KIO) and MTBD-High-MT complex DTT(+) (EMD-9997, PDB-6KIQ).

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

## Acknowledgements

This work was supported in part by grants for the Development of Core Technologies for Innovative Drug Development Based Upon IT, obtained from the Ministry of Economy, Trade and Industry (METI) and the Japan Agency for Medical Research and Development (AMED); by Grants-in-Aid for Scientific Research on Priority Areas (Grant Number 26119005 to N.N., and Grant Number 21121002 to I.S.) from the Japanese Ministry of Education, Culture, Sports, Science and Technology (MEXT); by a JST CREST grant to M.K. (Grant No. JPMJCR14M1); and by the Platform Project for Supporting Drug Discovery and Life Science Research (Basis for Supporting Innovative Drug Discovery and Life Science Research (BINDS)) from AMED under Grant Number JP19am01011115 (to M.K.).

## Author contributions

N.N., I.S., and M.K. conceived the project. N.N., O.T., A.W., S.T., and I.S. carried out the protein production and NMR experiments. N.N., Y.K., and M.K. performed the cryo-EM analyses. All authors discussed the data. N.N., Y.K., O.T., I.S., and M.K. wrote the paper.

## Competing interests

The authors declare no competing interests.
