## [Peer Review File · Nature Communications]

Reviewers' Comments:

Reviewer #1:

Remarks to the Author:

In this manuscript, Nishida et al. combine NMR and cryo-EM studies to provide new insights into the structure of the microtubule (MT)-binding domain (MTBD) of the MT-associated motor protein, cytoplasmic dynein, when bound to MTs. Specifically, the authors used a minimal truncated version of the MTBD-stalk coiled-coil domain of yeast dynein with the stalk helices cross-linked either in the weak MT-binding "+ β " registry (named here "MTBD-High") or the strong MT-binding " α " registry ("MTBD-Low"). Using NMR, the authors first solved the structures of both constructs, which confirmed the known composition of the MTBD of six α -helices (H1 to H6) and an anti-parallel coiled-coil (CC1 and CC2). Interestingly, while the registry of CC1-CC2 of the MTBD-Low construct was similar to the previously reported registry of the MTBD-stalk coiled-coil of mouse dynein in the + β registry, the registry of the MTBD-High construct was different from the expected α registry by an α -helical half-turn, which the authors coined the "semi- α " registration. Nishida et al. then used cryo-EM to visualize the conformations of the helices H1-H6 and the CC1-CC2 of the MTBD-High construct bound to MTs and found that the MT-bound MTBD-High construct assumes the α registration (named the "MTBD-Bound" structure). Here, CC1 changed by a full α -helical turn from the position of CC1 in the MTBD-Low construct. Based on these findings and the observation that the structure of the MT-bound MTBD-High construct remains unchanged when DDT is added (to cleave disulfide bonds), Nishida and co-workers suggest that MT-binding induces the α registration. The study will help to solve the current controversy surrounding the structure of the weakly and strongly MT-bound MTBD of dynein and should be of interest to the cytoskeletal motor community. I recommend publication by Nature Communications, subject to the following changes/additions:

1. Introduction, page 3: The authors may want to add at the end of "In addition, another coiled-coil structure, called the strut or buttress..." that the linker controls the conformation of the strut (Rao et al. 2019 Nat. Commun.).
2. Page 4, first paragraph: The authors write that "(3) The motor domain searches for a new binding site on the MT, via a weak interaction mode in either the ATP or ADP/Pi state. (4) The phosphate release induces strong binding to the MT, and the linker conformation reverts from bent to straight form in the MT-bound state to transmit the powerstroke". While the recent work by Rao et al. agrees with the statement that the initial MT binding of the leading motor domain occurs in the weak MT-binding + β registry, it also revealed that the strong MT-binding α registry can only be assumed when the linker is docked to AAA5, i.e. when the linker is in the post-powerstroke state. This suggests that phosphate release alone does not induce stronger binding to the MT and instead induces the power stroke, which in turn induces the strong MT-binding α registry. The author should modify their statements correspondingly or at least mention the findings by Rao and co-workers.
3. Page 5, first paragraph: The authors write that "It has been proposed that the switching of the MT affinity of MTBD is achieved by changes in the association mode of the coiled-coil...". As it has been demonstrated by three groups that registry changes result in affinity changes, I suggest replacing "proposed" by "shown".
4. Page 5, first paragraph: The authors write that "Biochemical studies have demonstrated that the sliding of CC1 with respect to CC2 by one-turn of an α -helix results in a distinctive change in the affinity for MTs". The authors may also reference here the single-molecule study by Rao et al., which not only provided insights into how coiled-coil registry changes result in MT-affinity changes but also revealed for the first-time which registrations the dynein stalk assumes when a dynein motor domain assumes the leading or trailing positions on the MT.
5. Page 5, first paragraph: The authors write that "Comparison of the crystal structures of the

motor domain in the ADP/Vi- and ADP-bound states have indicated that the change in the registry is caused by an altered interaction between the stalk and strut/buttress". As the structure-function studies by Rao et al. have also revealed that the strut/buttress mediates the transition from the β registry (unbound head) to the α registry (leading head) and the newly discovered γ registry (trailing head before ATP binding to AAA1), the authors may also want to mention that study here.

6. Results, page 7: The authors write that "The intramolecular disulfide bond formation in each of the two mutant proteins was confirmed by mobility shifts of the proteins when separated by SDS-PAGE analysis (Supplementary Fig. 1a) and by the appearance of a single set of peaks in the HSQC (heteronuclear single quantum coherence) spectra (Fig. 1f)". While the HSQC study may suggest that the cross-linking efficiency is close to 100%, the SDS-PAGE mobility shift analysis suggests that the cross-linking efficiency is only ~70-80% for the MTBD-Low construct. The authors should discuss this discrepancy and determine the cross-linking efficiencies for both constructs as done by Kon et al. (NSMB 2009). If the cross-linking efficiencies are indeed significantly less than 100% for MTBD-Low and possibly for MTBD-High, the authors should discuss how the NMR and cryo-EM structure determination was performed in the presence of possible subpopulations of MTBDs with different registrations.

7. Page 9, first paragraph: The authors write that "We refer to the registry of MTBD-High as "semi- α ", since the sliding of CC1 places that helix in an intermediate position between the + β - and α -registries". Is the "semi- α " registration exactly a half registry in between the + β - and α -registries? If so, the authors should call this registry the γ registry as this intermediate registration has been previously discovered and named the γ registry (Rao et al.).

8. Page 10: The authors performed cryo-EM studies in the absence and presence of DTT (to prevent disulfide bond formation and/or to cleave disulfide bonds) and find that the DTT(-) and DDT(+) structures of the MT-bound MTBD-High construct are very similar. It is not clear to what degree DTT cleaves the cross-linking of the stalk helices at the point of the preparation of the cryo-EM experiments as it appears that the DDT was only present during the cryo-EM studies but not during the purification of the construct (during which the disulfide bonds already form). This could explain why the DTT(-) and DDT(+) structures are similar. The authors should at least determine the percentage of stalk cross-linking in the presence of DTT using their mobility shift assay to rule out that a significant fraction of the stalk helices is still crosslinked.

9. Discussion , page 16: I recommend adding "following linker detachment" after "In the ATP-bound states, the stalk coiled-coil adopts the + β registry across the entire region" as dynein only assumes the + β registry after the linker detaches from AAA5 (see Rao et al.).

10. Page 16, Discussion: The authors write that "...which alters the interaction mode between the stalk and strut coiled-coils to induce the release of the stalk coiled-coil association mode from the + β -registry". The word "mode" is confusing in this sentence and the recent work by Rao et al. has provided additional insights into this pathway that the authors could mention here. I therefore recommend replacing this sentence with, for example, "which alters the interaction between the stalk and strut coiled-coils to induce the strong MT-binding α registration following the powerstroke (Rao et al)".

11. Page 16, Discussion: The authors write that "The complex formation with MTs stabilizes the conformation of MTBD in the α -registry, a change that is propagated through the entire coiled-coil helix toward the ATPase domain (Fig. 6b (4-3)). Consequently, further conformational rearrangements in the ATPase domain induce the powerstroke of the linker (Fig. 6b (5)) and the release of ADP (Fig. 6b (1))" and "...the conformational changes are transmitted toward the ATPase domain to facilitate ATP hydrolysis, release of Pi, and the powerstroke of the linker, all as a consequence of MT binding". Also here, the power stroke must occur before the α -registry can be induced. See also my comments #1, #2, #9 and #10.

12. Page 17: The idea that the dynein head initially binds the MT in the weak binding β registry finds direct support in the work by Rao and co-workers. The authors might therefore want to cite this work.

13. Finally, as the strong MT-binding α -registry is only assumed after the linker docks to AAA5 (post-powerstroke state), the authors should modify the schematics in Figures 6 and 7 correspondingly.

14. Methods, line 365: The authors should specify which concentration the MTBD solution had.

15. Finally, the methods section should be expanded to include how the authors expressed and purified all proteins as well as how the cryo-EM grids were prepared (used solutions etc.).

Reviewer #2:

Remarks to the Author:

Dynein, the ATP-driven microtubule-based motor, performs many critical functions in cells, including roles in intracellular transport, cell migration and cell division. Dynein is a large and complex motor, and there are several outstanding questions concerning the mechanism of its microtubule-stimulated ATPase and of its ATP-driven microtubule-based motility. The dynein ATPase site is separated by $\sim 15\text{nm}$ from its small, globular microtubule binding domain (MTBD), which sits at the end of a long, anti-parallel coiled-coil protruding from the body of the AAA dynein motor domain. Allosteric communication between these regions is central to dynein function but is not well understood.

Important context for the work of Nishida et al includes: 1) the study of Gibbons et al (2005) who investigated the stalk and MTBD out of the context of the extremely large dynein holoenzyme; by engineering these dynein elements within seryl-tRNA synthetase (SRS), they altered the register of the dynein coiled coil and thereby showed that this was important for coupling between microtubule binding/release; 2) Kon et al (2009) used cysteine cross links to trap the coiled coil in 3 distinct registrations, thereby modulating microtubule affinity and the sensitivity of the motor ATPase to microtubule binding; 3) building on determination of the X-ray structure of the dynein MTBD by Carter et al (2008), Redwine et al (2012) calculated a $\sim 10\text{\AA}$ resolution cryo-EM reconstruction of the microtubule-bound SRS-MTBD that showed how changes in microtubule affinity in the MTBD are correlated with structural changes within this domain; 4) Lacey et al (2019) also used cryo-EM to study the SRS-constrained stalk+MTBDs of both cytoplasmic and axonemal dynein 7, allowing them to visualise at higher resolution ($\sim 5\text{\AA}$) different microtubule-induced conformational changes in the microtubule-bound MTBDs compared to Redwine et al. Nishida et al appropriately cite these previous studies, but these available data also potentially limit the general interest of the work in the current manuscript.

Nishida et al combine solution NMR and cryo-electron microscopy (cryo-EM) studies to structurally investigate the behaviour of cytoplasmic dynein coiled-coil+MTBD constructs in isolation rather than located within an SRS context. Instead, the effect of constraining the register of the coiled-coil using disulphide cross-links on MTBD solution structure, microtubule binding affinity and the microtubule-bound structure are presented. The authors use "MTBD-Low" and "MTBD-High" constructs, which capture low and high microtubule affinity states respectively of the MTBD via differently positioned disulphides, and have been previously partially characterized by the authors using NMR (Takarada et al, *Biomol NMR Assign* 2014).

Nishida et al's NMR analyses provide additional evidence of the modulation of the structure of the MTBD by the registration of the coiled-coil. They determine the cryo-EM reconstructions of MTBD-High in the presence and absence of reducing agent, both of which adopt essentially the same

structure, at least at the presented resolutions of the reconstructions. While their cryo-EM analysis largely confirms the findings of Lacey et al., their work provides further evidence of the plasticity of the dynein MTBD. MTBD-High adopts the same conformation on the microtubule regardless of the coiled coil constraints, showing the importance of the structural communication between microtubule binding and the rest of the motor via the coiled coil. This would predict that reduced (unconstrained) MTBD-Low would also adopt this conformation, but this reconstruction is not included in the current work.

Nishida et al argue that their resolution is better than that of Lacey et al, but visual comparisons of these structures suggest that the differences are rather marginal. The mode of presentation by Nishida et al of the density in wire mesh format makes the details harder to discern. Further, at these resolutions, the 2nd decimal place of the resolution estimate is meaningless.

One striking aspect of the cryo-EM sample preparation methods is that MTBD-decorated microtubules for MTBD-High \pm DTT were prepared in different ways: oxidized MTBD-High was bound to prepolymerized microtubules, whereas reduced MTBD-High was bound to tubulin and then microtubules were polymerised in the presence of MTBD-High. Why was this necessary? Since it is increasingly well established that tubulin is subject to subtle conformational changes during polymerization, and the authors argue that the same is true for the dynein MTBD interaction with microtubules, how can the authors exclude that these different sample preparation modes are affecting the conformations determined and the conclusions drawn? This is particularly important to explain given that the register of the coiled coil is completely unconstrained under these conditions, and are therefore very different from that in the dynein holoenzyme.

Reviewer #3:

Remarks to the Author:

Dynein is an important motor protein whose power-producing mechanism is still elusive. The construct the authors produced are quite elaborate and impressive as they implemented multiple coiled-coil registries by using disulfide bonding within MTBD. The results appear quite excellent and solid with near atomic resolution EM complex data with MT combined with solution structure of multiple states MTBD, but the discussion may not be.

P16 L280 "This alternative pathway explains"

This argument should require more detailed explanation. It appears to the reviewer that ATPase-driven pathway also explains the enhancement of ATPase activity. So, how can only the MT-binding induced pathway explain the enhancement of the ATPase activity?

P17 L292 "the stalk would tilt backward"

It appears that the authors did not assume the elasticity of the stalk coiled-coil. As many papers have reported, the stalk is highly flexible. No one exactly knows how stiff the stalk is. So, the authors should mention the limitation of this model according to the possible insufficient strength of the stalk to induce such signal transduction that the authors assume.

Minor comments.

The figure numbers of P12 L194 and L195 do not make sense.

Reviewer #1 (Remarks to the Author):

In this manuscript, Nishida et al. combine NMR and cryo-EM studies to provide new insights into the structure of the microtubule (MT)-binding domain (MTBD) of the MT-associated motor protein, cytoplasmic dynein, when bound to MTs. Specifically, the authors used a minimal truncated version of the MTBD-stalk coiled-coil domain of yeast dynein with the stalk helices cross-linked either in the weak MT-binding “+ β ” registry (named here “MTBD-High”) or the strong MT-binding “ α ” registry (“MTBD-Low”). Using NMR, the authors first solved the structures of both constructs, which confirmed the known composition of the MTBD of six α -helices (H1 to H6) and an anti-parallel coiled-coil (CC1 and CC2). Interestingly, while the registry of CC1-CC2 of the MTBD-Low construct was similar to the previously reported registry of the MTBD-stalk coiled-coil of mouse dynein in the + β registry, the registry of the MTBD-High construct was different from the expected α registry by an α -helical half-turn, which the authors coined the “semi- α ” registration. Nishida et al. then used cryo-EM to visualize the conformations of the helices H1-H6 and the CC1-CC2 of the MTBD-High construct bound to MTs and found that the MT-bound MTBD-High construct assumes the α registration (named the “MTBD-Bound” structure). Here, CC1 changed by a full α -helical turn from the position of CC1 in the MTBD-Low construct. Based on these findings and the observation that the structure of the MT-bound MTBD-High construct remains unchanged when DDT is added (to cleave disulfide bonds), Nishida and co-workers suggest that MT-binding induces the α registration. The study will help to solve the current controversy surrounding the structure of the weakly and strongly MT-bound MTBD of dynein and should be of interest to the cytoskeletal motor community. I recommend publication by Nature Communications, subject to the following changes/additions:

1. Introduction, page 3: The authors may want to add at the end of “In addition, another coiled-coil structure, called the strut or buttress...” that the linker controls the conformation of the strut (Rao et al. 2019 Nat. Commun.).

According to the reviewer’s suggestions, we added the statement that the linker domain controls the conformation of the strut with citation to Rao et al 2019 Nat Commun (ref #8) at Page 3 and Line 17..

2. Page 4, first paragraph: The authors write that “(3) The motor domain searches for a new binding site on the MT, via a weak interaction mode in either the ATP or ADP/Pi state. (4) The phosphate release induces strong binding to the MT, and the linker conformation reverts from bent to straight form in the MT-bound state to transmit the powerstroke”. While the recent work by Rao et al.

agrees with the statement that the initial MT binding of the leading motor domain occurs in the weak MT-binding + β registry, it also revealed that the strong MT-binding α registry can only be assumed when the linker is docked to AAA5, i.e. when the linker is in the post-powerstroke state. This suggests that phosphate release alone does not induce stronger binding to the MT and instead induces the power stroke, which in turn induces the strong MT-binding α registry. The author should modify their statements correspondingly or at least mention the findings by Rao and co-workers.

According to the reviewer's suggestions, we cited the paper of Rao et al at the beginning of this paragraph. In this paragraph, we describe the relationship between the MT-binding affinity and ATP cycles in the ATPase domain. In addition, we introduce the coiled-coil registry of dynein stalk in following paragraphs (page 4-5) and discuss how the registry changes correlate with entire mechanocycles of dynein in Discussion (page 17, please see the comments #10 as well).

3. Page 5, first paragraph: The authors write that "It has been proposed that the switching of the MT affinity of MTBD is achieved by changes in the association mode of the coiled-coil...". As it has been demonstrated by three groups that registry changes result in affinity changes, I suggest replacing "proposed" by "shown".

Thanks to the comment, we replace "proposed" by shown (Page 5 and Line 1).

4. Page 5, first paragraph: The authors write that "Biochemical studies have demonstrated that the sliding of CCI with respect to CC2 by one-turn of an α -helix results in a distinctive change in the affinity for MTs". The authors may also reference here the single-molecule study by Rao et al., which not only provided insights into how coiled-coil registry changes result in MT-affinity changes but also revealed for the first-time which registrations the dynein stalk assumes when a dynein motor domain assumes the leading or trailing positions on the MT.

According to the reviewer's suggestions, we added a sentence that describes the single-molecule study performed by Rao et al.as follows at Page 5 and Line 4.

"A recent single-molecule study further demonstrated that the sliding of coiled coil helices and the resulting changes in interaction with stalk and strut/buttress regulates the MT-binding affinity and dynein motility."

5. Page 5, first paragraph: The authors write that "Comparison of the crystal structures of the

motor domain in the ADP/Vi- and ADP-bound states have indicated that the change in the registry is caused by an altered interaction between the stalk and strut/buttress". As the structure-function studies by Rao et al. have also revealed that the strut/buttress mediates the transition from the β registry (unbound head) to the α registry (leading head) and the newly discovered γ registry (trailing head before ATP binding to AAA1), the authors may also want to mention that study here.

In this sentence, we refer to the differences of the stalk conformation caused by strut/buttress interactions at atomic level. We described the functional consequence of stalk/strut interactions are described as a response for comment #4, with citation of the paper by Rao et al. We discuss the γ registry in the discussion (also see the response to comment #7)

6. Results, page 7: The authors write that "The intramolecular disulfide bond formation in each of the two mutant proteins was confirmed by mobility shifts of the proteins when separated by SDS-PAGE analysis (Supplementary Fig. 1a) and by the appearance of a single set of peaks in the HSQC (heteronuclear single quantum coherence) spectra (Fig. 1f)". While the HSQC study may suggest that the cross-linking efficiency is close to 100%, the SDS-PAGE mobility shift analysis suggests that the cross-linking efficiency is only ~70-80% for the MTBD-Low construct. The authors should discuss this discrepancy and determine the cross-linking efficiencies for both constructs as done by Kon et al. (NSMB 2009). If the cross-linking efficiencies are indeed significantly less than 100% for MTBD-Low and possibly for MTBD-High, the authors should discuss how the NMR and cryo-EM structure determination was performed in the presence of possible subpopulations of MTBDs with different registrations.

The engineered disulfide bond of MTBD-High and MTBD-Low was spontaneously formed during protein expression and purification process without oxidant treatment. We re-tested the MTBD-High and MTBD-Low samples with and without DTT reduction, followed by IAA treatment to block the free thiol group. As shown in the revised supplementary Fig.1a, the MTBD without DTT reduction, showed a single band with the mobility shift, indicating that the efficiency of S-S bond formation is nearly 100%, consistent with the results of HSQC spectra.

For NMR structure determination, we collected NOE distance information based on the signals of unambiguously assigned as MTBD-High and MTBD-Low by series of triple resonance NMR experiments. Therefore, we can obtain the structure of the disulfide bonded MTBD even if small amount of uncrosslinked form is present.

For cryo-EM analyses, we treated MTBD-High with 1 mM DTT for 2 hours at room temperature prior to EM grid preparation. As shown in Supplementary Fig 1f, the S-S bond is mostly reduced at this condition. Co-sedimentation assay demonstrated that the MT-binding affinity is mostly comparable with or without disulfide bond formation (i.e. between MTBD-High and MTBD-WT), indicating that the MTBD in complex with MT on the EM grid should mostly represents the disulfide reduced form.

7. Page 9, first paragraph: The authors write that “We refer to the registry of MTBD-High as “semi- α ”, since the sliding of CCI places that helix in an intermediate position between the $+\beta$ - and α -registries”. Is the “semi- α ” registration exactly a half registry in between the $+\beta$ - and α -registries? If so, the authors should call this registry the γ registry as this intermediate registration has been previously discovered and named the γ registry (Rao et al.).

There are several differences between the semi- α and γ registry. First, the semi- α conformation is observed only in the absence of MTs, and shifted to the α registry in the MT-bound state. In contrast, the γ registry persists in the MT-bound state, when the force was applied from either forward or backward direction. Second, the position of disulfide cross-linking is also different (semi- α is proximal to MTBD and γ registry is in the middle of stalk). Although we do not completely rule out the possibility that the semi- α and γ registry represents the same conformation in the entire dynein stalk coiled coil context, the structural analyses are necessary using the construct in which the disulfide bond is introduced at the γ registry position. We stated these points in the Discussion (Page 17 and Line 7).

8. Page 10: The authors performed cryo-EM studies in the absence and presence of DTT (to prevent disulfide bond formation and/or to cleave disulfide bonds) and find that the DTT(-) and DDT(+) structures of the MT-bound MTBD-High construct are very similar. It is not clear to what degree DTT cleaves the cross-linking of the stalk helices at the point of the preparation of the cryo-EM experiments as it appears that the DDT was only present during the cryo-EM studies but not during the purification of the construct (during which the disulfide bonds already form). This could explain why the DTT(-) and DDT(+) structures are similar. The authors should at least determine the percentage of stalk cross-linking in the presence of DTT using their mobility shift assay to rule out that a significant fraction of the stalk helices is still crosslinked.

We tested if the disulfide bond of MTBD-High is reduced in the presence of 1 mM DTT (i.e.

the same DTT concentration as EM grid preparation). As supplementary Fig. 1f shows, nearly 100% of the disulfide bond is reduced at 2 hours of DTT processing. We added the description about this experiment in the Method (Page 23 Line 13).

9. Discussion , page 16: I recommend adding “following linker detachment” after “In the ATP-bound states, the stalk coiled-coil adopts the + β registry across the entire region” as dynein only assumes the + β registry after the linker detaches from AAA5 (see Rao et al.).

We corrected the sentence according to the reviewer’s suggestion at Page 15 and Line 2.

10. Page 16, Discussion: The authors write that “...which alters the interaction mode between the stalk and strut coiled-coils to induce the release of the stalk coiled-coil association mode from the + β -registry”. The word “mode” is confusing in this sentence and the recent work by Rao et al. has provided additional insights into this pathway that the authors could mention here. I therefore recommend replacing this sentence with, for example, “which alters the interaction between the stalk and strut coiled-coils to induce the strong MT-binding a registration following the powerstroke (Rao et al)”.

The previous FRET assay using the single head dynein by Kon et al demonstrated that the phosphate release occurs before the powerstroke of linker. This should be necessary to ensure that the dynein motor achieve collective forward movements. Although we do understand the complete stabilization of the α registry of entire coiled coil stalk is induced after detachment of the linker from AAA5 (powerstroke), the registry of coiled coil may adopt conformational equilibrium of multiple states, and those equilibrium differs locally (i.e. near the MTBD or near the ATPase domain). In this discussion, we focus on the registry change near the MTBD. Upon release of Pi, the MTBD shifts the conformation toward the semi- α registry, which shows stronger MT-binding. The MT-binding stabilizes the registry near the MTBD into the α registry, triggering the subsequent conformational changes of ATPase domain, including the linker docking of AAA5 (powerstroke). As a result, the **entire** stalk region would be stabilized into the α registry, as shown by Rao et al. To reflect this point, we also revised the Fig. 6b.

11. Page 16, Discussion: The authors write that “The complex formation with MTs stabilizes the conformation of MTBD in the α -registry, a change that is propagated through the entire coiled-coil helix toward the ATPase domain (Fig. 6b (4-3)). Consequently, further conformational rearrangements in the ATPase domain induce the powerstroke of the linker (Fig. 6b (5)) and the

release of ADP (Fig. 6b (1))” and “...the conformational changes are transmitted toward the ATPase domain to facilitate ATP hydrolysis, release of Pi, and the powerstroke of the linker, all as a consequence of MT binding”. Also here, the power stroke must occur before the α -registry can be induced. See also my comments #1, #2, #9 and #10.

Please see comments #10

12. Page 17: The idea that the dynein head initially binds the MT in the weak binding β registry finds direct support in the work by Rao and co-workers. The authors might therefore want to cite this work.

According to reviewer's suggestion, we cited the paper of Rao et al at Page 18 Line 9.

13. Finally, as the strong MT-binding α -registry is only assumed after the linker docks to AAA5 (post-powerstroke state), the authors should modify the schematics in Figures 6 and 7 correspondingly.

We modified Fig.6b (step2, step4-2 to 4-3, and step 4').

2 The ATP-binding induce detachment of the linker from AAA5, and the change of stalk into the β registry.

4-2 Here, we only discuss the registry near the MTBD. Upon release of Pi, the MTBD exhibits the shifts the conformation toward the semi- α registry.

4-3 The MT-binding stabilizes the registry near the MTBD into the α registry, triggering the subsequent conformational changes of ATPase domain, including the linker docking of AAA5 (powerstroke). As a result, the entire stalk region stabilized into the α registry (5).

4' As same as above, we only describe the registry near the MTBD. The MT-binding induce the α registry only near the MTBD. Subsequently, this conformational change is transmitted to ATPase domain, and then induces multiple consequences; ATP hydrolysis, the powerstroke, and complete formation of the α registry throughout the entire stalk.

See also comments for #10

14. Methods, line 365: The authors should specify which concentration the MTBD solution had.

We added the information about MTBD solution for cryo-EM sample preparation at Page 23 and Line 13 -15.

15. Finally, the methods section should be expanded to include how the authors expressed and purified all proteins as well as how the cryo-EM grids were prepared (used solutions etc.).

We described the methods for protein expression and purification, and the EM grid preparation in the Method section.

Reviewer #2 (Remarks to the Author):

Dynein, the ATP-driven microtubule-based motor, performs many critical functions in cells, including roles in intracellular transport, cell migration and cell division. Dynein is a large and complex motor, and there are several outstanding questions concerning the mechanism of its microtubule-stimulated ATPase and of its ATP-driven microtubule-based motility. The dynein ATPase site is separated by ~15nm from its small, globular microtubule binding domain (MTBD), which sits at the end of a long, anti-parallel coiled-coil protruding from the body of the AAA dynein motor domain. Allosteric communication between these regions is central to dynein function but is not well understood.

Important context for the work of Nishida et al includes: 1) the study of Gibbons et al (2005) who investigated the stalk and MTBD out of the context of the extremely large dynein holoenzyme; by engineering these dynein elements within seryl-tRNA synthetase (SRS), they altered the register of the dynein coiled coil and thereby showed that this was important for coupling between microtubule binding/release; 2) Kon et al (2009) used cysteine cross links to trap the coiled coil in 3 distinct registrations, thereby modulating microtubule affinity and the sensitivity of the motor ATPase to microtubule binding; 3) building on determination of the X-ray structure of the dynein MTBD by Carter et al (2008), Redwine et al (2012) calculated a ~10Å resolution cryo-EM reconstruction of the microtubule-bound SRS-MTBD that showed how changes in microtubule affinity in the MTBD are correlated with structural changes within this domain; 4) Lacey et al (2019) also used cryo-EM to study the SRS-constrained stalk+MTBDs of both cytoplasmic and axonemal dynein 7, allowing them to visualise at higher resolution (~5Å) different microtubule-induced conformational changes in the microtubule-bound MTBDs compared to Redwine et al. Nishida et al appropriately cite these previous studies, but these available data also potentially limit the general interest of the work in the current manuscript.

As indicated by the reviewer, our cryo-EM model of MTBD-High in complex with MT is very similar to the one by Lacey et al (2019). However, we also analyzed the NMR structures of MTBD-High and MTBD-Low in the absence of MTs, and the cryo-EM structure of

MTBD-High in complex with MTs under the disulfide bond-reduced condition. Combination of these data, for the first time, demonstrated that the α -registry conformation of MTBD is induced by MT-binding, not by the engineered disulfide bond. Therefore, we think this finding is novel, and provided the conceptual advance for understanding of the mechanochemical cycles of cytoplasmic dynein.

Nishida et al combine solution NMR and cryo-electron microscopy (cryo-EM) studies to structurally investigate the behaviour of cytoplasmic dynein coiled-coil+MTBD constructs in isolation rather than located within an SRS context. Instead, the effect of constraining the register of the coiled-coil using disulphide cross-links on MTBD solution structure, microtubule binding affinity and the microtubule-bound structure are presented. The authors use “MTBD-Low” and “MTBD-High” constructs, which capture low and high microtubule affinity states respectively of the MTBD via differently positioned disulphides, and have been previously partially characterized by the authors using NMR (Takarada et al, *Biomol NMR Assign* 2014).

Nishida et al’s NMR analyses provide additional evidence of the modulation of the structure of the MTBD by the registration of the coiled-coil. They determine the cryo-EM reconstructions of MTBD-High in the presence and absence of reducing agent, both of which adopt essentially the same structure, at least at the presented resolutions of the reconstructions. While their cryo-EM analysis largely confirms the findings of Lacey et al., their work provides further evidence of the plasticity of the dynein MTBD. MTBD-High adopts the same conformation on the microtubule regardless of the coiled coil constraints, showing the importance of the structural communication between microtubule binding and the rest of the motor via the coiled coil. This would predict that reduced (unconstrained) MTBD-Low would also adopt this conformation, but this reconstruction is not included in the current work.

Thank you for appreciating the importance of our work that shows the plasticity of dynein MTBD. On the other hand, we did not perform the experiment the reviewer proposed in the last sentence (cryo-EM analysis of reduced MTBD-Low in complex with microtubule). This is because the MTBD-Low is the same as the MTBD-High construct except for the position of the cysteine of CC1 (S3097C for MTBD-Low and I3101C for MTBD-High) and we think that the comparisons among MTBD-High (constrained & in solution), MTBD-High (constrained & with MT) and MTBD-High (unconstrained & with MT) is enough to claim the effect of microtubule binding.

Nishida et al argue that their resolution is better than that of Lacey et al, but visual comparisons of

these structures suggest that the differences are rather marginal. The mode of presentation by Nishida et al of the density in wire mesh format makes the details harder to discern.

Comparing the local resolution of MTBD region between ours and that of Lacey et al. (4-5 Å vs 5-7 Å), we believe that the difference in resolution is significant. We changed the mode of presentation of the density in Fig. 2e and f by changing the width of the wire mesh and the way of lighting.

Further, at these resolutions, the 2nd decimal place of the resolution estimate is meaningless.

As suggested, we removed 2nd decimal from the resolution estimates and changed the resolutions of +/- DTT MTBD-High-MT complex to 3.7 Å and 3.9 Å.

One striking aspect of the cryo-EM sample preparation methods is that MTBD-decorated microtubules for MTBD-High ±DTT were prepared in different ways: oxidized MTBD-High was bound to prepolymerized microtubules, whereas reduced MTBD-High was bound to tubulin and then microtubules were polymerised in the presence of MTBD-High. Why was this necessary? Since it is increasingly well established that tubulin is subject to subtle conformational changes during polymerization, and the authors argue that the same is true for the dynein MTBD interaction with microtubules, how can the authors exclude that these different sample preparation modes are affecting the conformations determined and the conclusions drawn? This is particularly important to explain given that the register of the coiled coil is completely unconstrained under these conditions, and are therefore very different from that in the dynein holoenzyme.

We reviewed our lab notebook and found that the previous description of cryo-EM grid preparation of DTT(+) MTBD-High-MT complex was incorrect. MTBD-High was actually added to GMPCPP-MT solution after the MT polymerization. As MTBD-High easily aggregate at 37 degrees, it is at least not possible to make MT copolymerize with MTBD-High at such temperature. The corresponding method section was corrected accordingly.

Reviewer #3 (Remarks to the Author):

Dynein is an important motor protein whose power-producing mechanism is still elusive. The construct the authors produced are quite elaborate and impressive as they implemented multiple coiled-coil registries by using disulfide bonding within MTBD. The results appear quite excellent

and solid with near atomic resolution EM complex data with MT combined with solution structure of multiple states MTBD, but the discussion may not be.

P16 L280 “This alternative pathway explains”

This argument should require more detailed explanation. It appears to the reviewer that ATPase-driven pathway also explains the enhancement of ATPase activity. So, how can only the MT-binding induced pathway explain the enhancement of the ATPase activity?

The ATPase driven model cannot explain why the MT binding enhances the ATPase activity, since ATP in AAA1 is hydrolyzed before MTBD binds to MTs. In this study, we demonstrated that MTs binding alone can induce the α registry conformation in the CC1 and CC2 near the MTBD. This conformational change will be propagated to ATPase domain, which presumably facilitates the ATP hydrolysis of AAA1. To make clearer this point, we modified the Discussion section at Page 16 Line 15.

P17 L292 “the stalk would tilt backward”

It appears that the authors did not assume the elasticity of the stalk coiled-coil. As many papers have reported, the stalk is highly flexible. No one exactly knows how stiff the stalk is. So, the authors should mention the limitation of this model according to the possible insufficient strength of the stalk to induce such signal transduction that the authors assume.

We agree with the reviewer's suggestion. We added the description regarding the stiffness of the stalk is required to support this model at Page 17 Line 11.

Minor comments.

The figure numbers of P12 L194 and L195 do not make sense.

Corrected accordingly.

Reviewers' Comments:

Reviewer #1:

Remarks to the Author:

The revision has significantly strengthened the manuscript and satisfactorily answered my comments. I recommend publication subject to the following minor changes:

Line 277 and following: As Rao et al. (Nat. Commun. 2019) have shown that the stalk helices only assume the α -registry when the linker is docked to AAA5, the statement "Consequently, further conformational rearrangements in the ATPase domain induce the detachment of the linker to adopt entire stalk in the α -registry and the subsequent powerstroke of the linker (Fig. 6b (5))" is incorrect. The authors should therefore correct this statement.

Finally, the paper should be edited by a native English speaker to improve for readability and to correct grammar mistakes.

Reviewer #2:

Remarks to the Author:

The edits that have been made to the manuscript have substantially improved it and I am satisfied with the way in which the reviewers' comments have been addressed. Congratulations to the authors on this nice piece of work.

Reviewer #3:

Remarks to the Author:

I recommend the publication of this manuscript.

Point-by-point response to referees

Reviewer #1 (Remarks to the Author):

The revision has significantly strengthened the manuscript and satisfactorily answered my comments. I recommend publication subject to the following minor changes:

Line 277 and following: As Rao et al. (Nat. Commun. 2019) have shown that the stalk helices only assume the α -registry when the linker is docked to AAA5, the statement “Consequently, further conformational rearrangements in the ATPase domain induce the detachment of the linker to adopt entire stalk in the α -registry and the subsequent powerstroke of the linker (Fig. 6b (5))” is incorrect. The authors should therefore correct this statement.

I appreciate for the reviewer to point out the incorrect statement. We corrected the sentence as below;

Consequently, further conformational rearrangements in the ATPase domain induce the powerstroke of the linker (docked to AAA5) and a shift of the entire stalk in the α -registry (Fig. 6b (5)).

Finally, the paper should be edited by a native English speaker to improve for readability and to correct grammatical mistakes.

The manuscript was already subjected to professional proofreading service prior to the initial submission. According to the reviewer's suggestion, the revised manuscript was edited by the professional proofreading service again to correct the grammatical mistakes.

Reviewer #2 (Remarks to the Author):

The edits that have been made to the manuscript have substantially improved it and I am satisfied with the way in which the reviewers' comments have been addressed. Congratulations to the authors on this nice piece of work.

Reviewer #3 (Remarks to the Author):

I recommend the publication of this manuscript.